comparative functional genomics; plant evolution; plant–microbe interactions; symbiosis; symbiotic nitrogen fixation.

**Authors for correspondence:**
Sophie de Vries
E-mail: sophie.devries@uni-goettingen.de
Jan de Vries
E-mail: devries.jan@uni-goettingen.de

# Evolutionary genomic insights into cyanobacterial symbioses in plants

Sophie de Vries[1,,*] and Jan de Vries[1,,2,,3,,*]

[1]Department of Applied Bioinformatics, Institute for Microbiology and Genetics, University of Goettingen, Goettingen, Germany; [2]Goettingen Center for Molecular Biosciences (GZMB), University of Goettingen, Goettingen, Germany; [3]Campus Institute Data Science (CIDAS), University of Goettingen, Goettingen, Germany

### Abstract

Photosynthesis, the ability to fix atmospheric carbon dioxide, was acquired by eukaryotes through symbiosis: the plastids of plants and algae resulted from a cyanobacterial symbiosis that commenced more than 1.5 billion years ago and has chartered a unique evolutionary path. This resulted in the evolutionary origin of plants and algae. Some extant land plants have recruited additional biochemical aid from symbiotic cyanobacteria; these plants associate with filamentous cyanobacteria that fix atmospheric nitrogen. Examples of such interactions can be found in select species from across all major lineages of land plants. The recent rise in genomic and transcriptomic data has provided new insights into the molecular foundation of these interactions. Furthermore, the hornwort *Anthoceros* has emerged as a model system for the molecular biology of cyanobacteria–plant interactions. Here, we review these developments driven by high-throughput data and pinpoint their power to yield general patterns across these diverse symbioses.

## 1. Cyanobacterial symbioses in land plants—an overview

Chloroplasts emerged as a result of a cyanobacterial symbiosis. From a single endosymbiotic event about 1.5 bya (Bengtson et al., 2017; Butterfield, 2000; Eme et al., 2014), almost the entire diversity of photosynthetic eukaryotes radiated (Archibald, 2015). The nature of that cyanobacterial plastid progenitor is still debated (de Vries & Archibald, 2017; Ponce-Toledo et al., 2017); while some analyses suggest that the ancestor of a recently discovered unicellular cyanobacterium was the plastid progenitor (Ponce-Toledo et al., 2017) other studies suggest that this ancestral free-living, symbiotically competent cyanobacterium might have been a filamentous cyanobacterium—as these analyses recover affiliation to a section of extant filamentous cyanobacteria based on phylogenetic and sequence similarity data (Dagan et al., 2013; Ochoa de Alda et al., 2014). Up until today, this section of filamentous cyanobacteria brings about symbiotically competent strains that are the foundation of diverse cyanobacterial symbioses—from fungal to plant hosts—that we find on our planet (Rai et al., 2000). In these interactions, the cyanobacteria provide fixed nitrogen to their hosts.

Cyanobacterial symbionts (cyanobionts) of plants are present throughout the plant kingdom, yet they have emerged multiple times independently (Bergman et al., 1996; Rai et al., 2000; Figure 1). As a result, the degree of intimacy and the solutions on how the symbionts are housed are as different as the plants that undergo these symbioses (Figure 1): cyanobacterial symbioses of bryophytes are found among representatives from all major lineages, that is, mosses, liverworts and hornworts (Adams et al., 2013; Adams & Duggan, 2008). Feathermosses form epiphytic associations with cyanobionts. These are not 'accidental' associations, but the epiphytes are recruited and they fix nitrogen for the mosses (Bay et al., 2013; Stuart et al., 2020). In contrast, hornworts and some liverworts (e.g., *Blasia*) have special mucilage-filled cavities, auricles or canals (hornwort *Leiosporoceros dussii*) to host cyanobacteria (Adams et al., 2013; Adams & Duggan, 2008). Other liverworts have only epiphytic associations (Adams & Duggan, 2008); yet they are not as well studied as those from feathermosses (Bay et al., 2013; Ininbergs et al., 2011; Warshan et al., 2017). Cycads (gymnosperms) induce the formation of a special organ type, the coralloid roots, that are colonised intercellularly by cyanobionts via broken tissue or

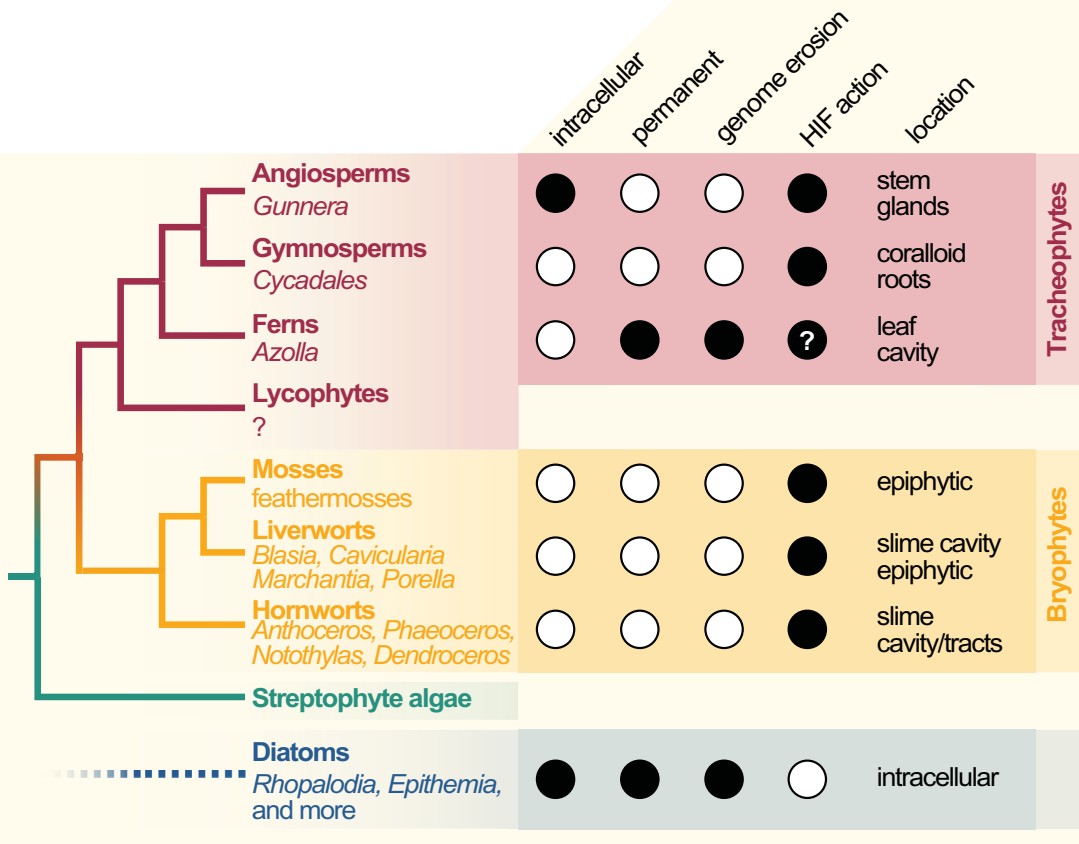

**Figure 1.** Cladogram of symbiotic interactions across land plants and diatoms. To the left, a simplified cladogram of the phylogenetic relationship among streptophytes (based on Puttick et al., 2018) is shown; diatoms (algae with a plastid of red algal origin) are very distantly related to streptophytes—the most recent common ancestor they share might be the last common ancestor of all eukaryotes (LECA). Among all major lineages of land plants except lycophytes (bold font), a few species that engage in symbiosis with cyanobacteria can be found (written in regular and italic font). To the right, filled dots denote key traits of the cyanobionts that dwell within those plants (and diatoms).

lenticels (Bell-Doyon et al., 2020; Rai et al., 2000; Reinke, 1872; Suárez-Moo et al., 2019). The angiosperm *Gunnera* attracts its cyanobionts through mucilage-filled channels into their stems, where they enter and become intracellular (Khamar et al., 2010; Nilsson et al., 2006; Towata, 1985).

All aforementioned symbioses are facultative. In contrast to them stands the fern genus *Azolla* with their cyanobionts. Here, cyanobionts are transferred from one generation to the next (de Vries & de Vries, 2018; Perkins & Peters, 1993; Zheng et al., 2009a). It is the only known permanent and obligate cyanobacterial symbiosis in the plant kingdom to date—that is apart from the chloroplast. Yet, in contrast to the chloroplast, the cyanobionts of *Azolla* are not endophytes, but live in special mucilage-filled cavities located in the dorsal leaf lobes (Figure 2).

Cyanobacterial symbioses do not occur in well-established model systems, although some of their hosts are currently emerging as such (Frangedakis et al., 2021; Li et al., 2018). That said, the availability, the accessibility and the recent improvements to large-scale data have given the exploration of cyanobacterial symbioses a new boost (Li et al., 2018; 2020). Genomics, transcriptomics, proteomics and metagenomics provide new insights into the evolutionary biology of cyanobacterial symbioses. They establish the foundation to explore common recurrent patterns and identify the unique solutions that the evolutionary distant host plants have evolved to establish nitrogen-fixing symbioses with cyanobacteria. Comparative genomics on the cyanobacterial partners have

identified patterns important in the biology of the bacteria, and those relevant for successful contributions to and gains from the symbiotic relationships. Here, we review the advances that the multiple different omics approaches have contributed to the understanding of cyanobacterial symbioses.

## 2. Recurrent molecular patterns in cyanobacterial symbioses: the cyanobionts

The scattered occurrences and diverse solutions in the establishment of the different plant-cyanobiont interactions may appear overwhelming. Yet, the questions that guide research on these symbioses are the same regardless of the symbiotic system. One of the key questions in plant-cyanobiont symbioses is what factors contribute to the initiation of the interaction. Here, much attention has been paid to the cyanobionts. Comparative genomics between symbiotically competent and not-competent cyanobacteria, as well as historically fingerprinting and nowadays metagenomic analyses, have been used to identify symbiotic communities and their features. Here, studies identified many recurring patterns.

### 2.1 Metagenomics, fingerprinting and cyanobacterial diversity

The question which cyanobacteria colonise the different hosts has led to several fingerprinting studies and isolation and Sanger sequencing of the cyanobacteria (e.g., Costa et al., 1999;

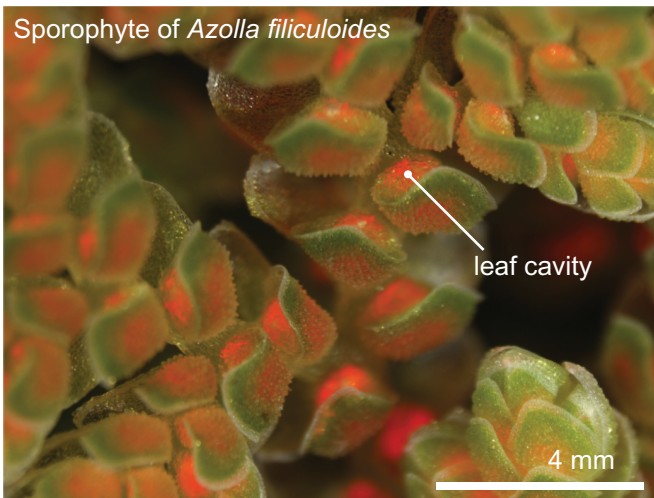

Sporophyte of *Azolla filiculoides*

leaf cavity

4 mm

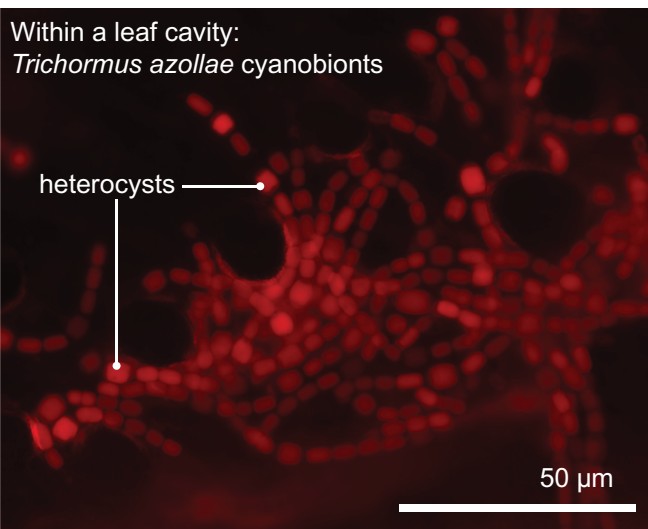

Within a leaf cavity:
*Trichormus azollae* cyanobionts

heterocysts

50 µm

**Figure 2.** The *Azolla–Trichormus* symbiosis. At the top, the sporophyte body of *Azolla filiculoides* whose leaf cavities are packed with cyanobionts (red autofluorescence) is shown. At the bottom, a fluorescence micrograph of the filamentous cyanobionts (*Trichormus azollae*) within one leaf cavity are shown (red autofluorescence); note the heterocysts, which are larger than the vegetative cells.

Ininbergs et al., 2011; Nilsson et al., 2000; Zheng et al., 1999). Independent of the host, section IV/V cyanobacteria, foremost those of the order Nostocales, are consistently found in cyanobacterial symbioses (e.g., Liaimer et al., 2016; Nelson et al., 2019; Pratte & Thiel, 2021; Ran et al., 2010; Warshan et al., 2018). This suggests some inert symbiotic property that comes with the cyanobacteria of these sections. Specifically, most of the cyanobionts have been identified either as *Nostoc* or *Anabaena* (Adams et al., 2013; Rai et al., 2000). The exception to the *Nostoc/Anabaena* symbionts is the cyanobiont of *Azolla*, which is as of late been called *Trichormus azollae* (see also discussions in Pereira & Vasconcelos, 2014). These symbiotic Nostocales are all filamentous, nitrogen-fixing and heterocyst-forming cyanobacteria. Some of them are able to colonise and associate in symbioses with various distinct hosts (Adams et al., 2013), suggesting that there may be limited host specificity.

Recent metagenomic data from hornworts uncovered that cyanobiont communities of the same host species from distinct locations can be quite different (Bouchard et al., 2020; Nelson et al., 2021). Location appears to play a major role in which

(cyano)bacterial community is established in the symbiotic space. Even in the hornwort *L. dussii*, which has specialised canals that run through the entire lobe, geographic location of the host is the major determinant of the cyanobiont community, suggesting that the ability to colonise those canals is not uncommon for symbiotically capable cyanobacteria co-occurring with *L. dussii* (Bouchard et al., 2020). What cyanobiont will colonise the symbiotic space of hornworts is likely determined by the community present in the soil (Nelson et al., 2021). Despite that, the soil harbours certain species that never colonise the host, suggesting that while soil type dictates what species can generally become cyanobionts, the host has some measure of selectivity to allow only certain species entry into its symbiotic space (Nelson et al., 2021). This appears to be a common theme for the microbiome in tissues with close contact to soil. Other metabarcoding studies have highlighted that soil type is the major determined of the bacterial and eukaryotic root microbiota of plants (Lundberg et al., 2012; Sapp et al., 2018; Schlaeppi et al., 2014). In contrast to observations in cycads and hornworts, high host specificity and less influence by environmental factors were observed for cyanobacterial communities associated with feathermosses (Ininbergs et al., 2011). Yet, in the lab, symbiotically competent *Nostoc* isolates from other hosts were generally capable to form a symbiotic association with these mosses (Warshan et al., 2017). Another symbiosis that differs in terms of specificity is that of *Azolla* with *T. azollae*. This symbiosis is obligate and exists for more than 60 my (Carrapiço, 2006; 2010; Collinson, 2002). The obligatory nature is likely due to the symbiont's highly reduced and eroded genome (Ran et al., 2010). Concomitantly, genomic sequencing and fingerprinting studies highlighted that this cyanobiont is co-evolving with its host (Li et al., 2018; Zheng et al., 1999).

### 2.2 Genomics of symbiotically competent cyanobacteria

Genomes of several symbiotic cyanobacteria have been sequenced. Most illuminating among these were the genomes of *obligate* symbionts. Data on *T. azolla* and the nitrogen-fixing cyanobionts of the diatoms *Rhopalodia gibba*, UCYN-A (a symbiont of a haptophyte) and *Epithemia turgida* all point towards genome erosion (Kneip et al., 2008; Nakayama et al., 2014; Ran et al., 2010). All of these genomes are in the process of being streamlined while maintaining genes required for nitrogen fixation ($N_2$-fixation). *Epithemia turgida* stands out by having completely eroded—or lost—genes for both photosystems.

In addition to the genome of *T. azollae* from *Azolla filiculoides* (Ran et al., 2010), cyanobiont genomes from five other *Azolla* species have been generated (Dijkhuizen et al., 2018; Li et al., 2018). The first genome of a cyanobacterial symbiont dwelling in plants was that of *Nostoc punctiforme* isolated from a cycad (Meeks et al., 2001). In contrast to that of *T. azollae* (5.49 Mb), its genome is 9.06 Mb and considered large for cyanobacteria (Meeks et al. 2001; Ran et al., 2010). After that genome sequences from nine isolates from feathermosses, *Blasia pusilla* and one nonsymbiotic *Nostoc* strain CALU 996 have been reported (Warshan et al., 2017; 2018), followed by 4 fully assembled genomes of isolates from *Blasia* and hornworts (Nelson et al., 2019), which are in the range of approximately 7 Mb in size, 6 isolates from a cycad (Gutiérrez-García et al., 2019) and 10 isolates of facultative symbionts from *Azolla* (Pratte & Thiel, 2021). For a comparison of genome sizes of cyanobacterial symbionts, see Figure 4a. In the next section, we will compare the findings gleaned from the different cyanobacterial genomes.

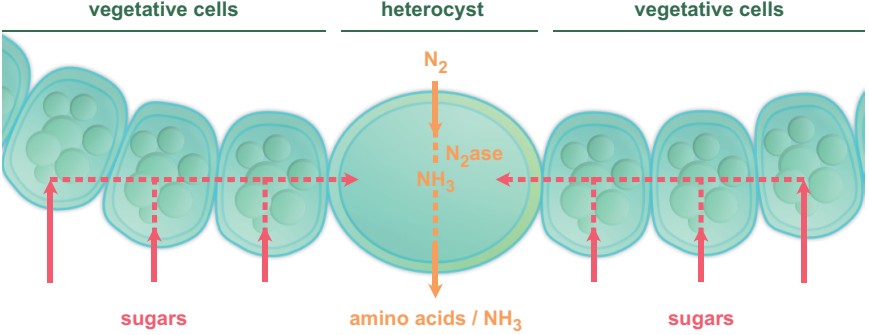

**Figure 3.** Syntrophic currencies exchanged by plant hosts and cyanobionts. A schematic Nostocales filament, consisting of vegetative cells and a heterocyst, is shown. The plant provides sugars (pink font), and the cyanobionts fix atmospheric nitrogen and provide it to the host via export (orange font). For an interesting discussion on putative transporters, see Roy et al. (2020).

## 2.3 Comparative genomics: cyanobionts versus nonsymbiotic cyanobacteria

What is special about certain cyanobacteria from the order Nostocales that they are recurrently associated in symbioses? Phylogenetic analysis showed that all analysed cyanobacteria that occur in plant symbioses are included in a larger monophyletic clade that, however, also contains nonsymbiotic cyanobacteria (Warshan et al., 2018). While *Nostoc* isolates that can associate intracellularly with *Gunnera* do form a monophyletic clade, extracellular and epiphytic isolates are polyphyletic within the larger clade of symbionts (Warshan et al., 2018). A study by Nelson et al. (2019), which sequenced genomes of additional isolates from bryophytes, corroborated this result. This further agrees with a phylogenomic survey across 650 diverse marine, freshwater and terrestrial cyanobacteria, that concluded that cyanobacteria with similar lifestyles tend to be more closely related to each other (Chen et al., 2021).

**2.3.1 Genomics of $N_2$-fixation.** The major contribution of the cyanobiont to the symbiosis with their plant hosts is fixed nitrogen. Filamentous cyanobacteria form specialised cells, called heterocysts, that have thicker cell walls, are photosynthetically inactive and create a microoxic environment inside the cell (Kumar et al., 2010; Figure 3). In this oxygen-poor environment, high levels of oxygen-sensible nitrogenase occur (see Gallon, 1981). In plant–cyanobacterial symbioses, heterocyst formation and the rate of $N_2$-fixation are increased; by how much depends on the respective host (Grilli Caiola et al., 1989; Hill, 1975; Wang et al., 2004).

$N_2$-fixing cyanobacteria encode two types of nitrogenase. One nitrogenase requires molybdenum (Mo) and iron (Fe) as co-factors and is expressed only in heterocysts (Mo-nitrogenase; Thiel et al., 1997). In the symbioses between *T. azollae* and *A. filiculoides,* it was hypothesised that Mo is provided and regulated by the host (Roy et al., 2020). This is based on the observation that the genome of *A. filiculoides* encodes a Mo transporter (Li et al., 2018), which, however, may likely be an importer for the plant (Roy et al., 2020). Its upregulation under nitrogen deprivation hints to an involvement in symbiosis (Eily et al., 2019). The Mo-nitrogenase is a complex that requires a range of enzymes for its synthesis and assembly (Dean et al., 1993). The complex itself requires the tetraheteromeric dinitrogenase encoded by *nifD* and *nifK* and the homodimeric dinitrogenase reductase encoded by *nifH*. Several nitrogen-fixing cyanobacteria encode an additional vanadium-based nitrogenase,

which also requires Fe (V-nitrogenase; Nelson et al., 2019; Pratte & Thiel, 2021; Thiel & Pratte, 2001). In *Anabaena variabilis* (now *N. variabilis*), transcription of the V-nitrogenase is repressed by both fixed nitrogen and the presence of Mo (Pratte et al., 2013). In the absence of Mo, V-nitrogenase activity is induced (Thiel, 1996). The gene cluster encoding V-nitrogenase and its regulators (Pratte et al., 2013), and the vanadate transporters (Pratte & Thiel, 2006), is conserved across cyanobacteria (Nelson et al., 2019). The first two assembled genomes from plant cyanobionts (*T. azollae* and *N. punctiforme* PCC 73102), however, likely lack V-nitrogenase-encoding genes (Meeks et al., 2001; Ran et al., 2010). Yet, recent studies (Nelson et al., 2019; Pratte & Thiel, 2021) report their presence in genomes of facultative, symbiotically competent cyanobacteria from *Azolla, Blasia* and hornworts. Nelson et al. (2019) reported the gene cluster in two of the cyanobionts they studied located on a plasmid and suggested the option for a gain via horizontal gene transfer. If and when V-nitrogenase plays a role under Mo-limited conditions during cyanobacteria–plant symbioses needs to be further investigated. A recent study on V-nitrogenase activity in lichen-associated cyanobacteria showed that availability of Mo correlated with the rate of $N_2$-fixation from V-nitrogenase (Darnajoux et al., 2019).

Most proteomic and transcriptomic approaches have focused on Mo-nitrogenase. Ekman et al. (2008) provided a proteomic comparison between the cyanobiont of the floating fern *A. filiculoides* and the symbiotic *Nostoc* isolate *N. punctiforme* PCC 73102 grown in culture. These data show that in the symbiotic association of *Azolla* proteins related to nitrogen metabolism were enhanced while those related to photosynthesis were reduced in the cyanobiont compared to the free-living, symbiotically capable *N. punctiforme* isolate. Similar tendencies were observed in the proteomic comparison of *N. punctiforme* PCC 73102 in symbiosis with *Gunnera manicata* and in culture (Ekman et al., 2006). In contrast, the epiphytic associations appear different. In feathermosses, nitrogen is transferred from cyanobiont to host (Bay et al., 2013), yet the expression of genes involved in $N_2$-fixation or heterocyst formation is not induced in symbiotic versus free-living *Nostoc* (Warshan et al., 2017). This suggests that the *Nostoc* isolates may have a similar rate of $N_2$-fixation in their symbiotic and free-living state. In agreement with that, both symbiotically competent *Nostoc* isolates analysed in this study identified *nifBHKU*—that is, genes encoding structural components nif H and K, a factor involved in nitrogenase assembly (nifU; Dos Santos et al., 2004; Johnson et al., 2005) and an enzyme required for the FeMo co-factor assembly (nifB; Curatti

et al., 2006)—and the regulator for heterocyst formation, *hetR,* in the top 10% of transcripts with the highest abundance in free-living and symbiotic condition (Warshan et al., 2017). In summary, while in close association the symbiosis appears to lead to an induction of $N_2$-fixation, the example of lose associations from the feathermosses suggests that here the host thrives already on the steady-state levels of fixed nitrogen provided by the cyanobionts. One could wonder whether this may be due to the specificity in symbiont recruitment described based on genomic fingerprinting of *nifH* sequences (Ininbergs et al., 2011); yet the observations were likewise true for a moss-specific and the symbiotically competent *N. punctiforme* isolate (Warshan et al., 2017).

What makes cyanobacteria symbiotically competent? And do they all use similar mechanisms for associating with their hosts? The latter appears likely given that most of them—with the exception of the obligate symbiont *T. azollae*–are able to colonise different hosts (e.g., Pratte & Thiel, 2021; Warshan et al., 2018). As such the question may rather be 'What are the molecular mechanisms of cyanobacterial symbioses?'. To answer these two questions, (comparative) genomic, transcriptomic and proteomic analyses have been carried out in different host-cyanobacteria associations.

### 2.3.2 Cyanobacterial symbioses from colonisation to communication during a functional symbiosis.

The ability to infect plants and associate in symbiosis is found in several members of the genus *Nostoc/Anabaena*. The last common ancestor of the monophyletic intra/extracellular symbiotic clade may have already been primed for the ability to associate with diverse hosts (Warshan et al., 2018). Yet this particular clade includes not only symbiotically competent isolates but also a variety of isolates not found in symbiosis. Warshan et al. (2018) provide a set of 170 gene families that are exclusively encoded in the genomes of 10 facultative *Nostoc* symbionts. These include, among others, genes coding for transporters involved in the import of organic sulphur compounds, phosphate and branched amino acids and ammonium (Warshan et al., 2018). Genomic data on cycad-derived symbionts found several metabolite biosynthesis pathways specific to the symbiotic cyanobacteria (Gutiérrez-García et al., 2019). Pratte and Thiel (2021) confirmed the presence of the majority of the gene families identified by Warshan et al. (2018) in the genomes of 10 *Nostoc* isolates that had previously been isolated from *Azolla* (Meeks et al., 1988; Zimmerman et al., 1989). These isolates had originally been obtained as the putative permanent and vertically transferred cyanobiont of *Azolla,* but were later identified as cyanobacteria that colonised the cavity in addition to *T. azollae* (Papaefthimiou et al., 2008). Infection assays with *Blasia* sp. confirmed that these isolates are capable of associating in symbioses and suggest a wider host range than only *Azolla* (Pratte & Thiel, 2021). However, genomic analyses of these symbiotic isolates and strains previously described as free-living identified several 'symbioses genes' in the free-living cyanobacteria. This may have two reasons: (a) Warshan et al. (2018) used one particular cyanobacterial strain as nonsymbiotic control, while Pratte and Thiel used a diverse set; and (b) symbiotic compatibility may be present even in free-living isolates that currently have not been found in metagenomic studies of symbiotic communities.

As described above, the genome of the symbiont of *Azolla* is smaller than that of most other symbiotically competent cyanobacteria (Meeks et al., 2001; Nelson et al., 2019; Ran et al., 2010; Figure 4a). For one, *T. azollae* does not belong to the genus *Nostoc* and has a distinct phylogenetic placement (Nelson et al.,

2019; Warshan et al., 2018). Moreover, the genome shows strong evidence of erosion (Ran et al., 2010). Many genes have been lost from the genome of *T. azollae* or subjected to pseudogenisation (Ran et al., 2010; Figure 4b,c). Accordingly, 28 gene families present in facultative symbionts have been lost in *T. azollae* (Warshan et al., 2018). Among those were genes involved in chemotaxis and in transport and metabolism of alkane and aliphatic sulfonate, underpinning the differences between obligate and facultative symbioses.

#### 2.3.2.1 Insights into motility and chemotaxis
What all cyanobionts have in common is their motility and ability to move towards their hosts. This is likewise true for facultative and obligate cyanobionts. In the case of *Azolla*, the filamentous motile stage—hormogonia—is induced to infect newly developed leaf cavities and sporocarps of the fern (Zheng et al., 2009b). Hosts can induce the formation of hormogonia—as is discussed later in the section on the hosts. Yet, the ability to move is not all; extracts of diverse hosts are able to attract cyanobacteria chemotactically (Nilsson et al., 2006). Genes involved in chemotaxis of hormogonia (*cheA, B, D, R* and *W*) are present in the genome of the symbiotically competent *N. punctiforme* (Meeks et al., 2001). The absence of functional CheR methyltransferase results in loss of motility and reduced colonisation of *N. punctiforme* of *Blasia* (Duggan et al., 2013). Furthermore, symbiotically competent strains induce chemotaxis-associated genes upon chemical contact with a putative host (Warshan et al., 2017). Accordingly, genes relevant for chemotaxis are among the 170 symbioses genes recovered by Warshan et al. (2018). In contrast, among the genes lost from the genome of *T. azollae* are some families involved in chemotaxis (Warshan et al., 2018). That is interesting because one would expect some host-derived guidance to the sporocarps or a new leaf cavity. But in contrast to facultative symbioses, movement of *T. azollae* may not only be guided by chemical communication. Trichomes are suggested to act as physical guidance towards new leaf cavities and sporocarps (Calvert et al., 1985; Calvert & Peters, 1981; Hill, 1989). Nonetheless, metabolic-based communication may be altered during the initiation of sporocarp formation. The transcriptional profile of transporter-encoding genes from *T. azollae* changed significantly during the induction of sporocarps (Dijkhuizen et al., 2021). Already early on in research on *Azolla*, it was suspected that host-derived phenolics and flavonoids in trichomes may be involved in the movement of *T. azollae* (Carrapiço & Tavares, 1989; Ishikura, 1982; Pereira & Carrapiço, 2007). Once the hosts—or new host tissue in case of *Azolla*—are colonised, hormogonia revert back to vegetative filaments (Bay et al., 2013; Meeks & Elhai, 2002; Zheng, Bergman, et al., 2009a). This repression is guided by the host and its regulation is governed by proteins encoded by the *hrm*-locus (Adams & Duggan, 2012; Campbell et al., 2003; Cohen & Meeks, 1997; Meeks, 1998).

The broad host range of many cyanobacterial symbionts that is occasionally observed in nature and used in the lab suggests that either (a) the symbioses rely on a common mechanism despite the diverse partners, (b) symbiotically competent *Nostoc* evolved the ability to recognise and respond a vast variety of different signals or (c) both.

#### 2.3.2.2 Insights into nutrient exchange and transport
An important aspect of cyanobacterial biology during symbioses is its transporter system. In symbioses, photosynthesis is reduced in cyanobionts as a trade-off for the higher rates of $N_2$-fixation (Bergman et al., 1992;

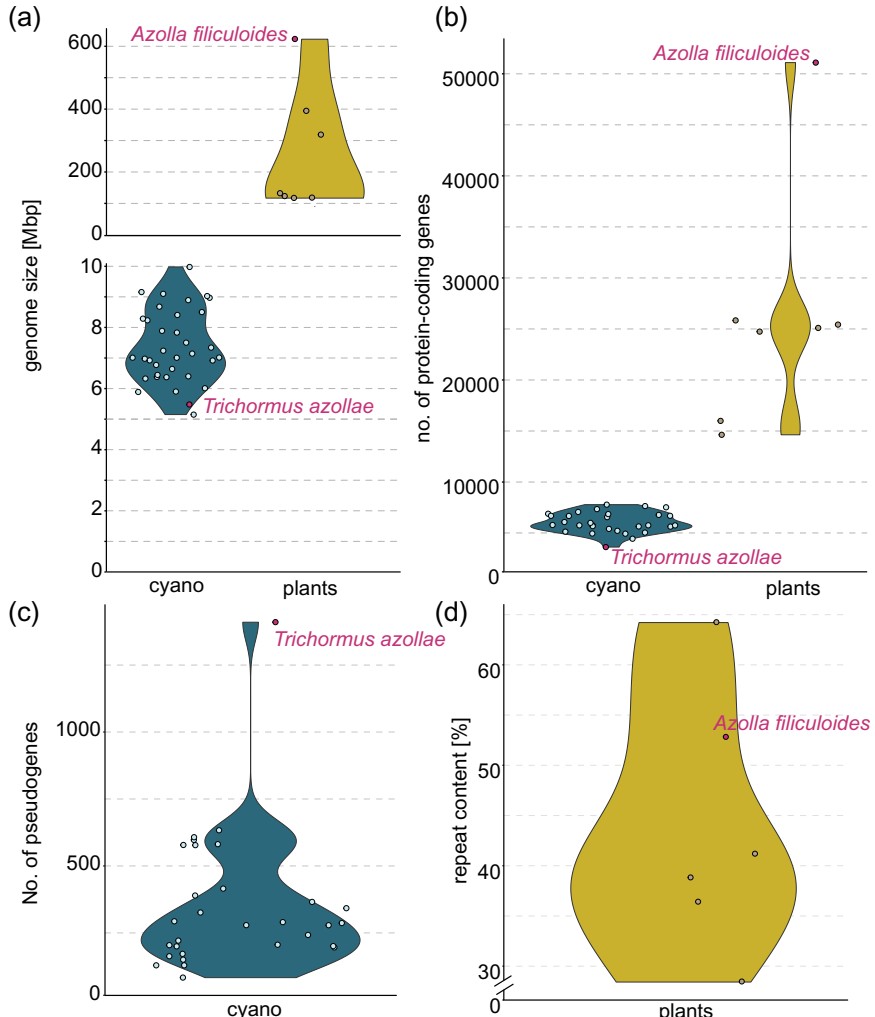

**Figure 4.** Genome characteristics of genomes from symbiotic and nonsymbiotic cyanobacteria and cyanobacterial host plants. (a) Genome size of several cyanobacteria shown to be symbiotic or free-living (blue, cyano) and all sequenced plant hosts for cyanobacteria (yellow, plants). (b) Number of annotated protein-coding genes in the genomes of cyanobacteria (blue, cyano) and cyanobacterial host plants (yellow, plants). (c) Number of pseudogenes described for the here included cyanobacterial genome data (for details, see description below). (d) Relative repeat content in the assembled plant genomes from plants symbiotically associating with cyanobacteria. To have a mixed set of cyanobacteria associating with plants and free-living, we included cyanobacteria previously sequenced and/or included in the comparative analyses by Pratte & Thiel (2021) and Nelson et al. (2019). NCBI accessions have been derived from these publications, and actual data have been obtained from the National Center for Biotechnology Information (NCBI). For cyanobacterial hosts, all host species that have been sequenced (*Anthoceros agrestis, Anthoceros punctatus, Anthoceros angustus, Azolla filiculoides, Sphagnum fallax* and *Pleurozium schreberi*) are included in the figure. The data are derived from Li et al. (2018, 2020), Pederson et al. (2019), Zhang et al. (2020), and Genome statistic information from Phytozome genome ID: 522. Data on repeat content were not reported for *S. fallax*, and the number of pseudogenes has not been reported for cyanobacteria isolated from feathermosses.

Ekman et al., 2008; Peters & Meeks, 1989; Rai et al., 1989; 2000). Thus, cyanobionts need to be provided with carbon from their hosts (Peters & Meeks, 1989; Rai et al., 2000; Söderbäck & Bergman, 1993; Stewart & Rodgers, 1977; Figure 3). Indeed, proteomics of *T. azollae* indicated an increase in a transporter associated—by sequence homology—with a phosphotransferase system fructose-specific IIC component (Ekman et al., 2008). In agreement, both in symbiosis with *Gunnera* and *Azolla*, key enzymes of the oxidative pentose phosphate pathway are induced (Ekman et al., 2006; 2008). This may suggest that fructose is the sugar transported between hosts and cyanobionts. However, a mutant study on sugar transporters of symbiotically competent *N. punctiforme* showed that only mutants where glucose permease function was impaired were unable to infect the hornwort *Anthoceros punctatus* (Ekman et al., 2013). This leaves four options: (a) cyanobionts do not receive the same type of sugar from every host system, (b) protein sequences that were by homology associated with

fructose transport and metabolism may need to be reinvestigated with new additional data at hand, (c) a lack of fructose during an established interaction may be circumvented, while glucose is required as a signal for the establishment of infection or (d) glucose is the transported sugar used as a carbon source for cyanobionts.

Warshan et al. (2018) found that the genomes of facultative symbiotic cyanobacteria are enriched in transporters for phosphate and organic sulphur and the metabolism of the latter. Both phosphate and sulphate transporters are induced upon contact between the feathermoss *Pleurozium schreberi* and symbiotic *Nostoc* isolates (Warshan et al., 2017). These data point to organic sulphur and phosphate as important nutrients delivered by the host. And indeed, sulphur compounds are a currency in feathermoss–cyanobiont symbioses (Stuart et al., 2020). Furthermore, elimination of a functional alkane sulfonate monooxygenase by targeted mutagenesis of symbiotic *N. punctiforme* showed that sulphur

metabolism is required for establishment of a symbiosis with feathermosses (Stuart et al., 2020).

***2.3.2.3 Insights into cyanobacterial stress during symbiosis*** Nitrogen starvation induces a stress response in cyanobacteria (Christman et al., 2011). It is thus not surprising that early proteomic analyses on the cyanobionts of *Azolla* and *Gunnera* identified an induction of proteins involved in stress (Ekman et al., 2006; 2008). In both host systems, these included proteins in oxidative stress. This agrees with data on cyanobacteria epiphytically associating with feathermosses. While both competent as well as non-competent *Nostoc* isolates highly expressed proteins involved in oxidative stress responses, some of these were only induced in the competent isolates upon physical contact with the mosses—both in the secreted proteome and transcriptome (Warshan et al., 2017). Moreover, these same proteins were reduced in the secreted protein fraction of the non-competent *Nostoc* isolate in this study. Additionally, in a metabolic screen, some *Nostoc* isolates, including some cyanobionts from *Blasia*, have been described to produce the cyanobacterial toxin microcystin (Liaimer et al., 2016). A microcystin deficient mutant showed higher sensitivity to reactive oxygen species (ROS; Zilliges et al., 2011), suggesting a protective role of these compounds against oxidative stress. Indeed, the microcystins were not secreted by symbiotic *Nostoc* isolates supporting a role inside the symbiont (Liaimer et al., 2016). Overall, these data suggest that a vital part of establishing and possibly maintaining $N_2$-fixing symbioses in many host-cyanobiont interactions relies on the induction of protection against ROS. This notion is corroborated by insights into a different nitrogen-fixing symbiosis: that occurring between rhizobial bacteria and legumes. A study by Günther et al. (2007) on *Lotus japonicus* has shown that high respiration rates mediated by leghemoglobins (which keep free $O_2$ levels low while acting in internal $O_2$ trafficking; see Appleby, 1984) lead to elevated levels of ROS.

### 2.4 Metagenomics and fingerprinting point to a bacterial microcosm surrounding the cyanobionts

No organism exists alone in its natural habitat. Already decades ago, the co-occurrence of other bacteria, next to cyanobionts, was noticed (Carrapiço, 1991; Wallace & Gates, 1986). Metagenome sequencing has uncovered a diversity of non-cyanobacterial symbionts (e.g., Dijkhuizen et al., 2018; Nelson et al., 2021). That cyanobionts are surrounded by various bacteria was noted for many systems. Thus, the host and cyanobiont do not only interact with each other, but with an entire microbial community. One might find this unsurprising. Bacteria and archaea are omnipresent, and a diversity of them associates with roots and leaves of plants. What roles they play and how essential the bacterial composition is to their plant hosts are, in general, the topics of current research (Fitzpatrick et al., 2020; Trivedi et al., 2020). The question though is, how are they capable to occupy the symbiotic space from the rather 'open' versions such as cavities of hornworts, liverworts and *Azolla* to more 'closed' options such as coralloid roots of cycads.

Are these bacteria capable of hijacking the symbiotic crosstalk between host and cyanobiont or do symbiotic spaces fulfil a general role in attracting symbionts for the plant? In a metagenomic study on hornwort cavities, it was observed that the nonsymbiotic associates follow the same patterns of specificity as the cyanobionts (Nelson et al., 2021). The original soil community dictates what organisms are present and as such have a chance to be recruited; here too, the diversity of the cavity does not equal that of the soil

per se. Hence, *Anthoceros* has some measure of control, which microbes it allows in and which it does not. In *Azolla* species, there are recurring members of the microbial community that are attuned to the developmental status of the symbiotic organs and the cyanobacterial physiology (Dijkhuizen et al., 2018). These bacteria were found to be co-transferred during vegetative and sexual transmission of the cyanobacterial population (Carrapiço, 1991). In contrast to the other symbioses, in *Gunnera petaloidea* and *Gunnera chilensis*, fungi and bacteria were observed within both the gland cells and the mucilage (Johansson & Bergman, 1992; Towata, 1985), but there are no records on the presence of such microbes in the exact same symbiotic spaces as the cyanobionts. Yet, these are only microscopic observations, and no molecular insights have been garnered.

These observations suggest a complex interaction between host, cyanobacteria and other microorganisms. The nature of these interactions is to a large degree not yet fully understood. With the omics approaches at hand, the nature of these interactions can now be further investigated. Dijkhuizen et al. (2018) performed the first comprehensive survey of the microbiome that occurs in six different *Azolla* species. The authors found that there are recurring Rhizobiales in the *Azolla* leaf pockets. Interestingly, similar presence of Rhizobiales has been reported in the endospheres of the hornwort *Leiosporoceros* (Bouchard et al., 2020), and the cycads *Dioon* and *Zamia* (Bell-Doyon et al., 2020; Suárez-Moo et al., 2019). While these recent microbiome studies suggest that Rhizobiales might have certain significances in plant–cyanobacteria symbiosis, their prevalence and functional roles remain to be tested. Additionally, Dijkhuizen et al. (2018) identified putative denitrifiers in the leave pocket of *Azolla,* suggesting that cheaters might exist in the symbiotic spaces. Yet, with the exception of the analyses of *Azolla*, no functional predictions have been made for the microbial community in cyanobacterial symbioses.

Currently, no comparative metagenomic study across the cyanobacterial hosts exists. Additionally, most metagenome sequencing studies focus on marker genes instead of entire genomes; this hampers inferences on functional roles in the symbiotic community and the identification of non-cyanobacterial symbionts or cheaters. Thus, how similar the microbiome in the symbiotic space of the phylogenetically distinct hosts is compared to nonsymbiotic tissues can only be approximated. Yet, because in all cyanobacterial hosts, facultative cyanobionts have some similar, symbiosis-associated genomic content and they perform similar functions within the symbiosis (Pratte & Thiel, 2021, Warshan et al., 2018), they might create a similar milieu in different symbiotic spaces. One could, hence, expect that the functional composition of the microbiome, be it beneficial or cheating, might be similar. Indeed, with more metagenomic data accumulating, it is becoming clear that while taxonomic compositions vary strongly, there are recurrent patterns in the functional composition of a microbiome in similar environments (Burke et al., 2011). After all, it is the biochemical properties of an environment on which selection acts (see excellent discussion in Doolittle & Inkpen, 2018). Future studies will illuminate convergence of functions in the symbiotic microverse across these diverse interactions.

## 3. Recurrent molecular patterns in cyanobacterial symbioses—a host-oriented perspective

The sequencing of cyanobacteria associated with nitrogen-fixing symbioses is for a long time on-going, and as a consequence, many

of the strains have been sequenced. For practical reasons regarding the much larger genomes (Figure 4a), from cost to computational power and the availability of methods to cope with a high repetitive content (Figure 4d), the sequencing of the host genomes has progressed more slowly. However, in the last few years, the genomes of several host plants have been sequenced: *Azolla filiculoides* (Li et al., 2018), four hornwort genomes [two *Anthoceros agrestis* ecotypes and *A. punctatus* (Li et al., 2020) and *Anthoceros angustus* (Zhang et al., 2020)] and the feathermoss *P. schreberi* (Pederson et al., 2019). Additionally, the 1KP project (Matasci et al., 2014) covers transcriptomes of *G. manicata,* 4 cycads (*Cycas micholoitzii, Stangeria eriopus, Dioon edule* and *Encephalartos barteri*), 10 different hornwort species from diverse genera, the liverwort *Blasia* sp. and an additional *Azolla* species. Additional transcriptomes for *A. filiculoides,* two other *Azolla* species, *Azolla pinnata* and *Azolla caroliniana,* and the hornwort *A. punctatus* exist (Brouwer et al., 2017; Chatterjee et al., 2021; de Vries et al., 2016; Li et al., 2020; Qi et al., 2018).

### 3.1. Establishment of interaction with cyanobacterial and other symbionts

Some of the now existing data were already available and included in a large comparative analysis that targeted the distribution and evolution of the symbiotic toolkit associated with arbuscular mycorrhizal (AM) symbioses (Radhakrishnan et al., 2020). The authors found that symbiotic genes involved in these particular symbioses are not present in land plants undergoing cyanobacterial but no AM symbioses. This suggests that even though the symbiotic pathways associated with the establishment of AM symbioses have been recruited for nitrogen-fixing symbioses with rhizobia (Parniske, 2008), cyanobacterial symbioses rely on different pathways for their interaction. Several data from *Gunnera,* cycads, hornworts, *Blasia* and feathermosses show that specific attractants exist (Bay et al., 2013; Campbell & Meeks, 1989; Khamar et al., 2010; Nilsson et al., 2006; Rasmussen et al., 1994), and that the presence of the hosts induces hormogonia formation (Bay et al., 2013; Campbell & Meeks, 1989; Meeks & Elhai, 2002; Ow et al., 1999; Rasmussen et al., 1994; Figure 1). Even in the interaction between *Azolla* and its cyanobiont, hormogonia formation and specific attraction to the sporocarps as well as the apex of the shoot (where new leaves are formed) is observed (Calvert et al., 1985; Hill, 1989; Perkins & Peters, 1993; Peters et al., 1978; Peters & Meeks, 1989; Zheng, Bergman, et al., 2009a).

Campbell and Meeks (1989) found a hormogonia inducing factor (HIF) of a size between 12 and 14 kDa in the hornwort *A. punctatus*. They did not further identify the nature of the secreted signal but suggested a poly-phenol-like molecule (Campbell & Meeks, 1989; Meeks & Elhai, 2002). Rasmussen et al. (1994) identified that mucilage of different *Gunnera* species induced hormogonia. Here, too, a molecule of about 12 kDa was found. In this case, proteinase K treatment suggested it to be a peptide. The accumulating transcriptomic and genomic data can now be helpful in identifying short peptides that may act as HIFs and test their function in the lab. Another HIF, a mixture of diacylglycerols, was recently identified from the cycad *Cycas revoluta* (Hashidoko et al., 2019). The major compound 1-palmitoyl-2-linoleoyl-*sn*-glycerol was the most active HIF. Additionally, sugars are in discussion as putative HIFs. Indeed, the secreted mucilage of mature glands of *Gunnera* and soluble sugars within it induced hormogonia formation (Khamar et al., 2010).

Because $N_2$-fixation is inhibited in hormogonia, a reversion to filaments with heterocysts, the $N_2$-fixing cells of cyanobacteria, is needed for the establishment of successful symbioses. This is triggered by all cyanobacterial symbioses (Meeks & Elhai, 2002). Glucose and fructose, which are almost absent from the mucilage of *Gunnera*, but accumulate within the gland tissues, repressed the formation of hormogonia (Khamar et al., 2010). Sugars have also been highlighted as putative attractants for the cyanobionts (Nilsson et al., 2006). Functional analyses of the genome of the symbiotic *N. punctiforme* showed that it encodes a functional fructose transporter, a glucose permease and an OprB family porin, which was associated with the uptake of both sugars (Ekman et al., 2013). In other cyanobacteria, this porin family has recently been associated with transport of different metals (Qiu et al., 2021; Schätzle et al., 2021). Additionally, some mutants of porin-like genes from *Anabaena* sp. PCC 7120 had altered phenotypes of the outer membrane, affected integration of other porins in the membrane or affected nitrogen demand and/or fixation (Schätzle et al., 2021). Hence, the OprB family appears to have a diversity of roles in bacterial biology; however, whether the role of OprB in sugar uptake is primary or a secondary effect cannot be distinguished because membrane integrity or integration of other proteins into the membrane was not assessed by Ekman et al. in 2013. In agreement with the data from Khamar et al. (2010), all mutants for the fructose and glucose transporters were able to form hormogonia (Ekman et al., 2013). Yet only mutants unable to produce the glucose permease resulted in the abortion of infection of the hornwort *A. punctatus* (Ekman et al., 2013). Given that the sugar content, including glucose, is depleted in *Gunnera* glands once successfully colonised by cyanobacteria (Khamar et al., 2010) suggests that glucose may additionally be supplemented by the plant as food for the symbiont.

### 3.2. Symbiotic communication and integration into stress responses

Transcriptomic analyses of the interaction between *A. filiculoides* and *T. azollae* recently highlighted a chalcone synthase-encoding gene (*CHS*) that has highly induced transcript levels only in the presence of the cyanobiont (Eily et al., 2019; Li et al., 2018), suggesting a commitment to flavonoid biosynthesis in symbiosis. And indeed, deoxyanthocyanins alone can increase the induction of the hormogonia-repressing factor *hrmA* by 70% and even more so in combination with naringenin, but not with other phenylpropanoid(-derived) compounds that were tested by Cohen et al. (2002). The inhibiting effect was also visible when extracts of leaf tissue from different *Azolla* species were used (Cohen et al., 2002). Agreeingly, several flavonoids were unable to promote hormogonia formation (Campbell & Meeks, 1989; Nilsson et al., 2006). Nonetheless, phenolics, including flavonoids, have been suggested to be involved in the transfer of cyanobionts to newly developed leaf cavities and sporocarps of *Azolla* (Pereira & Carrapiço, 2007). Expression of homologs of genes encoding enzymes in the phenylpropanoid pathway (see also de Vries et al., 2021b), providing the precursors for flavonoid biosynthesis, and flavonoid biosynthesis in *A. filiculoides*, is dependent on the circadian rhythm and the availability of nitrogen (Güngör et al., 2021). Anthocyanins produced by *Azolla* are elevated under various stresses, including protection against light stress (Nham Tran et al., 2020). It is no surprise that genes coding for enzymes involved in anthocyanin biosynthesis are responsive to various triggers and stresses. One of the intriguing questions is what role do they play in the symbiosis.

Given the permanent relationship of *Azolla* and its cyanobiont that needs to be maintained during stress, it is conceivable that stress-responsive metabolites, such as flavonoids, act as signals in the *Azolla* spp.–*Trichormus* symbioses. Proteomic analyses of salt stress on *Azolla microphylla* showed changes affecting the plant physiology and enzymes involved in $N_2$-fixation of the cyanobiont (Thagela et al., 2017; Yadav et al., 2019). The phytohormone salicylic acid (SA), which is involved in biotic stress in land plants (Pieterse et al., 2009), alters cyanobiont abundance and *nifE* gene expression (de Vries et al., 2018). Vice versa, transcriptomics and comparative genomic analyses highlight that the presence or absence of the cyanobiont alters the expression of homologous genes putatively involved in SA biosynthesis and signalling (de Vries et al., 2021a). More experimental and comparative data are needed to understand (a) how stress-responsive signalling integrates into *Azolla's* symbiosis and (b) whether this can be transferred to non-permanent cyanobacterial symbioses.

### 3.3. Nutrient exchange from a genomic host perspective

A recurrent theme when analysing large-scale data from cyanobacteria and their hosts is, as already exemplified above, the transport of sugars to the cyanobionts as a counter-currency of the symbioses (Figure 3). Indeed, carbon transport from feathermosses to their epiphytic cyanobionts was demonstrated (Stuart et al., 2020). Transcriptomic analyses of the hornworts *A. punctatus* and *A. agrestis* with and without cyanobiont highlighted a sugar transporter from the SWEET1 clade that increases its transcript level in a symbiotic state (Li et al., 2020). In the genome of *A. filiculoides*, 15 different SWEET genes have been identified, none of which can be linked to the symbioses based on transcriptomic data from cyanobiont-containing and cyanobiont-free cultures (Eily et al., 2019). That does not mean that they play no role in symbiosis in general; furthermore, other sugar transporters could be involved. This illustrates that the picture cannot be transferred between symbiotic systems in a 1:1 manner. Additionally, the case of *A. filiculoides* might be special because of the perpetuity of its symbiosis. Whether facultative associations between evolutionary distinct plants and their cyanobionts use similar mechanisms is to be tested. Yet, it is not unthinkable that convergent molecular mechanisms have emerged that tip into the same pathways, given that the cyanobacteria appear capable of infecting many of these possible hosts.

### 4. Conclusion

Cyanobacteria can occur as symbionts in a diversity of plants that are separated by millions of years of evolution. Despite this diversity, genomic data indicate that among the facultative symbionts, similar molecular mechanisms are used to interact with their hosts. This is in agreement with the ability of these cyanobionts to infect distinct host lineages. In contrast, transcriptomic data on the cyanobacteria and their hosts hint that every host system recruits unique molecular features particular to each individual association; this is in line with the convergent evolution of these cyanobacterial symbioses and the unique characters and different degrees of intimacy observed in each system. It is conceivable that symbiotically competent cyanobacteria must have some plasticity in their response to the different hosts and an ability to recognise each individual species. Comparative genomics and transcriptomics have the power to illuminate shared patterns across these diverse systems.

## Acknowledgement

We thank the anonymous reviewer for several valuable comments.

**Financial support.** J.d.V. thanks the European Research Council for funding under the European Union's Horizon 2020 research and innovation programme (Grant Agreement No. 852725; ERC-StG 'TerreStriAL'). J.d.V. is grateful for support through the German Research Foundation (DFG) within the framework of the Priority Programme 'MAdLand—Molecular Adaptation to Land: Plant Evolution to Change' (SPP 2237; VR 132/4-1).

**Conflicts of interest.** The authors declare no conflicts of interest.

**Authorship contributions.** S.d.V. and J.d.V. have outlined and written the manuscript together; both S.d.V. and J.d.V. created the figures.

**Data availability statement.** Data availability is not applicable to this article as no new data were created or analysed in this study.

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
