## [Reviewer Report]

Dear Dr. Hamant,

Thank you for the invitation to submit our Review article "Evolutionary genomic insights into cyanobacterial symbioses in plants" to Quantitative Plant Biology.

As previously outlined, we discuss recent developments in the field of plant–cyanobacteria symbiosis, which were driven by the generation of high-throughput-data. We pinpoint their power to yield general patterns across these diverse symbioses occurring in representative plant species from across all major lineages.

Yours sincerely

Sophie de Vries and Jan de Vries

---

## [Reviewer Report]

*Comments to Author*: This is a thorough review about symbioses of terrestrial plants with cyanobacteria. I have mostly minor comments, except for one serious complaint:

Lines 329/330 and Lines 504-510: you refer to Ekman et al. 2008 and Ekman et al. 2013 (which, by the way, are missing in your References section) about the function of proteins encoded by genes induced during symbiosis. At the time, the function of these proteins was assessed based on sequence similarity. Meanwhile, more protein functions have been analysed, see specifically Qiu et al. 2020 https://pubmed.ncbi.nlm.nih.gov/33196124/. Please check whether the definitions of protein function can be upheld, and adapt your discussion accordingly.

Line 58: ”(Reinke 1872).” – I like the historical depth, but one more reference that is (more easily) accessible would be appreciated.

Line 60: “(Towata 1985).” – and here, ditto.

Line 71: replace “emerge” with “emerging”

Line 75: “It sets the seed…” – should be “They”, not “It”, and “sets the seed” should be replaced by a more common phrase… lies the foundations, maybe?

Line 92: “the symbiotic communities and its features” – should be “their features”

Line 96: “The question on what cyanobacteria colonize…” – should be “The question which cyanobacteria colonize…”

Line 99: “…section IV/V cyanobacteria, foremost that of the order of Nostocales,…” – should be “foremost those of the order Nostocales”…

Line 114: Nelson et al. 2020 (and Nelson et al. 2019) is missing in the References section

Lines 117/118: “suggesting that no particular specialization is needed to colonize those canals” – This is a bit vague. The trait enabling the colonization of the canals may not be uncommon in some habitats, but it’s still a specialization.

Line 123: “(Nelson et al. 20210)” – missing in the References section (even after correction of the year)

Line 134: “the genome of T. azollae is obligate “ – replace "genome" with "symbiosis"

Lines 144/145: “…Nakayama et al., 2014)” – missing in the References section

Lines 152/153: “The first genome of a cyanobacterial symbiont of plants was sequenced by JGI and was an isolate of N. punctiforme from a cycad” – change of subject; please rephrase

Line 153, line 155: “(Meeks et akl. 2001)” – please correct

Lines 160/161: “we will set the findings from the different cyanobacterial genomes in comparison.” – “we will compare…” would be preferable

Lines 168/169: “While those Nostoc isolates that can associate ntracellular with Gunnera do form a monophyletic clade, extracellular and epiphytically” – intracellularly, extracellularly

Lines 187/188: “One nitrogenase requires molybdenum (Mo) as a co-factor and is expressed only in heterocysts” – It also requires iron.

Lines 197/198: “an additional vanadium-based nitrogenase (V-nitrogenase; Thiel and Pratte 2001, Nelson et al. 2019, Pratte and Thiel 2021).” – It also needs iron.

Lines 210/211: “As to if and when V-nitrogenase can play a role under Mo-limited conditions during symbioses needs to be further investigated.” – I recommend Darnajoux et al. 2019 https://www.pnas.org/content/116/49/24682

Lines 216/217: “These data show that proteins related to nitrogen metabolism were enhanced while those related to photosynthesis were reduced.” – you mean, in symbiosis?

Lines 221/222: “yet the expression of genes involved in nitrogen fixation or heterocysts formation are not induced…” – should be “…or heterocyst formation is not induced…”

Lines 223/224: ”This suggests a continuous rate of nitrogen fixation in contrast to an increased one in the tighter associations.” – better explain “continuous”, e.g., “a rate of nitrogen fixation similar to that observed in the free-living state”

Lines 227-229: “and an enzyme required for the iron-Mo co-factor assembly (nifB; Curatti et al. 2006—and the regulator for heterocysts formation, hetR,...” – It’s the FeMo cofactor. Close the parentheses. Heterocyst formation, not heterocysts formation

Line 235: “nifH sequences” – nifH in italics, please

Line 256: “These include, among other genes a large set of transporters…” – are you talking about genes of proteins?

Line 264: “as the putative permanent and inherited cyanobiont of Azolla” – why not use “vertically transferred” instead of “inherited”?

Lines 270/271: “(i) Warshan et al. (2018) used one particular cyanobacteria as non-symbiotic control,” – should be “cyanobacterium”. The singular is “cyanobacterium”. Or write “cyanobacterial strain”.

Line 277: “For one as its current name indicates T. azollae that it does not belong to the genus” – comma after “For one” and “indicates”, delete “that is”

Lines 292/293: “Genes involved in chemotaxis of hormogonia (CheA, B, D, R and W)” – do not capitalize cheA

Lines 294/295: “The absence of the functional CheR Methyltransferase” – do not capitalize methyltransferase

Lines 301/302: “to the sporocarps or a new leave cavity.” – leaf

Lines 311/312: “in case of Azolla—are colonized hormogonia revert back to” – comma after ”colonized”

Lines 329/330: “proteomics of the cyanobiont of Azolla indicated an increase of a fructose transporter (Ekman et al. 2008).” – The authors did not determine the protein function; they used sequence homology. I checked the protein sequence and found that nowadays, it shows the strongest homology to “iron uptake porin”.

Besides, the Ekman et al. papers (2008, 2013) are missing in the References section

Lines 336/337: “(ii) that a lack of fructose during an established interaction may be circumvented,” – delete ”that” for syntax reasons, and delete the entire point because the fructose transport function seems to be dead. Besides, the specificity of a sugar transporter for a _single_ monosaccharide should always be confirmed experimentally; sequence homology is not sufficient here.

Lines 347-350: “The gene families involved in transport and metabolism of organic sulfur, which are retained by the facultative symbionts, are lost from the reduced genome of T. azollae, the cyanobiont of Azolla (Ran et al. 2010; Warshan et al. 2018).” – you mentioned that already.

Lines 373-376: “Overall, these data suggest that a vital part of establishing and possibly maintaining nitrogen fixing symbioses—or the constitutive nitrogen fixing state in such symbioses—in many host-cyanobiont interactions relies on the induction of protection against reactive oxygen species.” – you could make a reference to legume nodules here – high respiration rates under microaerobic conditions, enabled by leghemoglobin, lead to increased ROS production during respiration, Günter et al. 2007 https://pubmed.ncbi.nlm.nih.gov/17990967/

Lines 384/385: ”That symbiotic cyanobacteria are surrounded by various bacteria has been noted for every system.” – Seriously? What about Gunnera symbioses?

Lines 392/393: “…to the more ‘closed’ options such as coralloid roots of cycads or the glands of Gunnera species.” – I still do not see any reference for non-cyanobacteria being found in infected cells of Gunnera glands. It’s not that I do not believe the possibility – there are non-rhizobia in infected cells of legume nodules, after all – but I want proof.

Lines 409-412: “such microbes have been observed in the same symbiotic spaces as the cyanobiont; i.e. in those specific gland cells that harbour the symbiotic Nostoc filaments. Yet, these are only microscopic observations-based data and no molecular insights have been garnered.” – I did not see any bacteria or fungi in the infected cells in Towata 1985. They were in the mucilage or cell walls - based on the figures and the figure legends.

Lines 477-479: “that is the formation of motile filaments, which are further specifically attracted to their hosts (Nilsson et al. 2006).” – The only new information about hormogonia here is that they are filaments. I think it would be better to have the full introduction at first occurrence in the text.

Lines 504-507: “Functional analyses of the genome of the symbiotic N. punctiforme has shown that it encodes a functional fructose transporter, a glucose permease as well as a carbohydrate porin which can transport both sugars (Ekman et al. 2013).” – the carbohydrate porin is now annotated as an iron porin. Please check the function of the proteins.

Lines 509-510: “Yet only mutants unable to produce the glucose permease resulted in the abortion of infection of the hornwort A. punctatus (Ekman et al. 2013).” – please check whether the glucose permease definition can be upheld.

Lines 557/558: “…That does not mean that they play no role in symbiosis in general. It however shows...” – There could be other sugar transporters involved.

Lines 558/559: “…shows that the picture cannot be transferred between the symbiotic systems in a 1:1 manner.” - you would only expect orthologous genes to be involved if the symbioses had a common evolutionary origin. Is there any reason to assume this?

Lines 559/560: ”... This may be for an instance because the relationship between A. filiculoides and its permament and obligate cyanobiont” – I do not know what you mean with “for an instance” here. 

Lines 573-576: “In contrast, transcriptomic data on the cyanobacterial and host hint on the fact that every host system— in line with the convergent evolution of these cyanobacterial symbioses and the unique characters and different degrees of intimacy observed in each system—highlight unique molecular features particular to each individual association.” – sentence does not compute; please rephrase

---

## [Reviewer Report]

*Comments to Author*: Dear Dr. de Vries

Thank you for your submission. I am sorry for the serious delay in our review process. 

We have contacted 16 reviewers and only 2 agreed to review. The first reviewer returned the review on time. But the second reviewer withdrew his/her consent of review after 3 weeks overdue. 

Since I do not want to keep you waiting, as it might take a very long time to look for another reviewer, after reading your manuscript and the first review, I think the review of the first reviewer is sufficient to further polish the article. 

Hence, please kindly amend your manuscript according to the recommendations of this reviewer. 

Sorry again and thank you very much for your patience. 

Yours sincerely

Boon Leong Lim

Associate Editor

---

## [Reviewer Report]

Dear Professor Hamant, dear Professor Lim

Thank you handling our manuscript. We are pleased that the reviewer found our work interesting and thorough.

In this revised version of our manuscript, we have made sure to tackle all of the constructive comments made by the reviewer. Please see our responses in the response letter, where we have labelled our responses with “>>>>AU:”.

Furthermore, we have added a completely new figure (Figure 4), that better captions quantitative aspects (coding content, genome sizes, pseudogenization etc.) of the sequenced diversity discussed here.

17. December: Additionally, we have now introduced the requested shortening, additional layout changes, and further integration of the figures into the text.

20. December: We have re-ordered the sections.

We remain

Yours sincerely

Sophie de Vries and Jan de Vries

---

## [Reviewer Report]

*Comments to Author*: We are sorry for the delay. We invited a second reviewer and today, one month after he accepted the invitation, he just recommended acceptance without submitting a detailed report . Thank you very much for revising this manuscript to its current